# Endothelial Dysfunction in Pulmonary Hypertension: Cause or Consequence?

**DOI:** 10.3390/biomedicines9010057

**Published:** 2021-01-09

**Authors:** Kondababu Kurakula, Valérie F. E. D. Smolders, Olga Tura-Ceide, J. Wouter Jukema, Paul H. A. Quax, Marie-José Goumans

**Affiliations:** 1Department of Cell and Chemical Biology, Laboratory for CardioVascular Cell Biology, Leiden University Medical Center, 2300 RC Leiden, The Netherlands; K.B.Kurakula@lumc.nl; 2Department of Surgery, Einthoven Laboratory for Experimental Vascular Medicine, Leiden University Medical Center, 2300 RC Leiden, The Netherlands; v.f.e.d.smolders@lumc.nl (V.F.E.D.S.); P.H.A.Quax@lumc.nl (P.H.A.Q.); 3Department of Pulmonary Medicine, Hospital Clínic-Institut d’Investigacions Biomèdiques August Pi i Sunyer (IDIBAPS), University of Barcelona, 08036 Barcelona, Spain; olgaturac@gmail.com; 4Department of Pulmonary Medicine, Dr. Josep Trueta University Hospital de Girona, Santa Caterina Hospital de Salt and the Girona Biomedical Research Institut (IDIBGI), 17190 Girona, Catalonia, Spain; 5Biomedical Research Networking Centre on Respiratory Diseases (CIBERES), 28029 Madrid, Spain; 6Department of Cardiology, Leiden University Medical Center, 2300 RC Leiden, The Netherlands; j.w.jukema@lumc.nl

**Keywords:** pulmonary hypertension, endothelial dysfunction, vasoactive factors, EndoMT, inflammation, TGF-β, epigenetics

## Abstract

Pulmonary arterial hypertension (PAH) is a rare, complex, and progressive disease that is characterized by the abnormal remodeling of the pulmonary arteries that leads to right ventricular failure and death. Although our understanding of the causes for abnormal vascular remodeling in PAH is limited, accumulating evidence indicates that endothelial cell (EC) dysfunction is one of the first triggers initiating this process. EC dysfunction leads to the activation of several cellular signalling pathways in the endothelium, resulting in the uncontrolled proliferation of ECs, pulmonary artery smooth muscle cells, and fibroblasts, and eventually leads to vascular remodelling and the occlusion of the pulmonary blood vessels. Other factors that are related to EC dysfunction in PAH are an increase in endothelial to mesenchymal transition, inflammation, apoptosis, and thrombus formation. In this review, we outline the latest advances on the role of EC dysfunction in PAH and other forms of pulmonary hypertension. We also elaborate on the molecular signals that orchestrate EC dysfunction in PAH. Understanding the role and mechanisms of EC dysfunction will unravel the therapeutic potential of targeting this process in PAH.

## 1. Introduction

Pulmonary hypertension (PH) is a condition that is defined by a mean pulmonary arterial pressure of more than 20 mmHg at rest and 30 mmHg during exercise. The range of genetic, molecular, and humoral causes that can lead to this increase in pressure is extensive. Therefore, PH is grouped into different classes that are based on clinical and pathological findings as well as therapeutic interventions [1,2]. The World Health Organization (WHO) classifies PH into five groups, namely: 1. Pulmonary arterial hypertension (PAH), 2. Pulmonary hypertension due to left heart disease (PH-LHD), 3. Pulmonary hypertension due to lung disease (PH-LD), 4. Chronic thromboembolic pulmonary hypertension (CTEPH), and 5. Pulmonary hypertension due to unclear and/or multifactorial mechanisms [1,3,4]. PH is increasingly becoming a global health issue due to the ageing population. Although PH-LHD and PH-LD are the most prevalent PH groups, research and drug development mainly focus on PAH and CTEPH, which are rarer diseases that mainly affect younger people [5]. This review will focus mostly on PAH because of the amount of research conducted in PAH as compared to the other four groups.

PAH is characterized by remodeling of distal pulmonary arteries, causing a progressive increase in vascular resistance. Vascular remodeling is associated with alterations in vasoconstriction, pulmonary artery- endothelial cells (PAECs) and -smooth muscle cells (PASMCs) cell proliferation, inflammation, apoptosis, angiogenesis, and thrombosis, which leads to the muscularization and occlusion of the lumen of pulmonary arteries by the formation of vascular lesions. Some of the lesions found in PAH are plexiform lesions, which are characterized by enhanced endothelial cell (EC) proliferation, thrombotic lesions and neointima formation, the formation of a layer of myofibroblasts, and extracellular matrix between the endothelium and the external elastic lamina [6,7]. One of the first triggers for development of PAH is thought to be EC injury triggering the activation of cellular signaling pathways that are not yet completely understood.

In normal conditions, the endothelium is in a quiescent and genetically stable state. When activated, the endothelium secretes different growth factors and cytokines that affect EC and SMC proliferation, apoptosis, coagulation, attract inflammatory cells, and/or affect vasoactivity in order to restore homeostasis. Prolonged or chronic activation of the endothelium leads to EC dysfunction, the loss of homeostatic functions, leading to pathological changes, and it is crucial in the development of cardiovascular diseases and so too in PAH [8,9]. Many different factors have been suggested to be triggers of EC dysfunction in PAH, like shear stress, hypoxia, inflammation, cilia length, and genetic factors (Figure 1) [6,10,11,12]. As a consequence, the endothelium switches from a quiescent to an overactive state, where it starts to secrete vasoconstrictive factors, like endothelin-1 (ET-1) [13] and thromboxane [14], and proliferative factors, like vascular endothelial growth factor (VEGF), fibroblast growth factor 2 (FGF2) [15], CXCL12 [16], and reduce the secretion of vasodilators, like nitric oxide (NO) and prostacyclin, which indicates that EC dysfunction might play a central role in the pathogenesis of PAH. Whether EC dysfunction is the primary cause or rather the consequence of changes in environmental factors remains to be resolved [8,17].

The purpose of this review is to provide a state-of-the-art overview on the features and driving forces of EC dysfunction in PAH and highlight the current progress made in understanding this phenomenon. Finally, this review discusses several models for studying EC dysfunction in PH and explores possible molecular targets and drugs for restoring EC function in PH.

## 2. Factors contributing to EC Dysfunction in PH

Approximately 80% of familial PAH (hPAH) and 20% of idiopathic cases of PAH (iPAH) are associated with mutations in the bone morphogenic type 2 receptor (BMPR2), but a penetrance of 20–30% suggests secondary stimuli, such as inflammation and thrombosis, as important contributors to EC dysfunction and PAH development [18,19,20,21]. More recently, alterations in endothelial metabolic functions in the pulmonary vasculature are emerging as important regulators of endothelial dysfunction.

### 2.1. Bone Morphogenic Type 2 Receptor

BMPR2 encodes for a transmembrane serine/threonine kinase receptor belonging to the transforming growth factor-β (TGFβ) family of signaling proteins (Figure 2) [22]. BMPR2 modulates cellular growth, apoptosis, inflammation, and differentiation via the binding of bone morphogenetic proteins (BMPs) to a heteromeric complex of a BMP type-I receptor and BMPR2, in a time, concentration, and cell type dependent manner [23]. BMPs are secreted cytokines that play important roles in vascular development and homeostasis. Alterations in the functions of BMPs are associated with severe developmental disorders and diverse human disease [23,24,25]. BMPR2 promotes the survival of PAECs depending on the localization in the vascular bed, and it has an anti-proliferative effect on PASMCs [26,27,28].

To date, over 380 PAH related mutations in *BMPR2* are known, mostly loss of function mutations [29,30]. The low penetrance of disease development associated with *BMPR2* mutations observed in humans has also been confirmed in experimental models of PH, where *BMPR2* deletion alone does not induce PAH in the majority of the cases [31,32,33]. Interestingly, reduced levels of BMPR2 have also been found in PH patients without *BMPR2* mutations, which suggests the additional involvement of genetic modifiers or environmental factors reducing BMPR2 dependent signaling [34,35,36,37].

BMPR2 is predominantly present in ECs lining the vascular lumen in the lung and expression is reduced in ECs from PH lung. Therefore, mutated *BMPR2* is postulated to play a significant role in EC dysfunction in PAH [30,34]. Association between endothelial BMPR2 expression levels and PAH development was further supported by the observation that mice with endothelial specific deletion of *BMPR2* were prone to developing PAH [38,39]. PAECs overexpressing a kinase-inactive BMPR2 mutant show increased susceptibility to apoptosis and conditioned medium from these PAECs stimulated proliferation of PASMCs via increased release of TGFβ1 and fibroblast growth factor (FGF)-2 [40]. More recently, mutations in *GDF2*, the gene encoding the BMP9 ligand, have been identified in PAH patients and associated with reduced circulating levels of both BMP9 and BMP10 [41]. The presence of these PAH-linked mutations in the endothelial BMPR2/ligand axis provide additional genetic evidence to support a critical role for endothelial dysfunction in the pathobiology of PAH. Moreover, BMP9 administration selectively enhanced endothelial BMPR2 signaling in PAECs and reversed PH in both MCT and SuHx rats [42]. Based on this knowledge, one might speculate a causal role for these mutations in EC dysfunction and subsequent PAH development.

Further evidence in the association between BMPR2 and EC dysfunction comes from studies showing that BMPR2 deficiency in iPAH PAECs is associated with the loss of DNA damage control via reduced DNA repair related genes, such as BRCA1 [43]. In addition, transcriptome analysis of PAECs from iPAH patients revealed a correlation between reduced BMPR2 levels and the downregulation of β-catenin, resulting in reduced Collagen-4 (COL4) and ephrinA1 (EFNA1) expression [44]. COL4 and EFNA1 both perform intertwining roles in endothelium structure. Moreover, siRNA mediated silencing of *BMPR2* in PAECs resulted in increased PAEC proliferation, migration, and the disruption of cytoskeletal architecture. One of the changes observed was an increase in Ras/Raf/ERK signaling, and Ras inhibitors, like nintedanib [45], reversed the enhanced proliferation and hypermotility of BMPR2 silencing in PAECs [46].

### 2.2. Inflammation

Mutations in *BMPR2* are known to predispose patients to developing PAH, but low penetrance and the time of disease onset suggest that a second hit required developing PAH. Pulmonary inflammation is such a plausible second hit that puts patients with BMPR2 mutations at risk of developing PAH. Exposure of *Bmpr2* mutant rats to 5-lipoxygenase, inducer of lung inflammation, induced severe PAH pathology with an endothelial transformation that required TGF-β signaling [47]. However, the administration of only IL-6 to rats and overexpression of IL-6 in transgenic mice also led to the occlusion of pulmonary arteries and RV hypertrophy without a silent mutation of BMPR2 [48,49]. Accordingly, it has also been found that pro-inflammatory cytokine TNFα in vitro downregulates the expression of BMPR2 via NOTCH signaling in ECs [50]. Altogether, this could suggest that sustained inflammation is an important trigger in PH development, potentially through the induction of EC dysfunction. Pulmonary arteries of PAH patients showed the infiltration of macrophages, dendritic cells, and lymphocytes into the plexiform lesions and an increased migration of monocytes [10,51]. Increased levels of pro-inflammatory cytokines and chemokines, such as IL-1β, TNFα, and IL-6, which are known activators of vascular endothelium, were found (Figure 3) [52,53,54]. Hence, has been found that IL-1β stimulates endothelial ET-1 production [55].

### 2.3. Thrombosis in PAH

The presence of thrombotic lesions in the pulmonary vasculature is a common pathological finding in PAH [56]. However, the role of thrombosis in PAH remains controversial. Few studies demonstrated that coagulation factors, such as proteases, tissue factor (TF), factor Xa, and thrombin, activate the coagulation cascade, which leads to the formation of fibrin clots that obstruct/narrow the lumen, could promote EC dysfunction, and can eventually contribute to vascular remodeling in PAH (Figure 3) [57]. In contrast, some studies support the hypothesis that thrombosis is an epiphenomenon of vascular remodeling in PAH [58]. Thus, it is still unknown whether thrombosis contributes to the pathogenesis of PAH or acts as a bystander.

Although altered platelet activation has been reported in PAH patients, their exact role in PAH remains controversial. Only platelets and ECs express and release von Willebrand Factor (vWF) upon activation, which facilitates the interaction between each other. Circulating vWF levels are significantly increased in PAH patients, which suggests the potential involvement of platelets in EC dysfunction in PAH [59]. CD40L, a proinflammatory mediator, is expressed on the surface of activated platelets. Upon activation, CD40L is cleaved into its soluble form (sCD40L), which is known to be greatly increased in PAH patients [60]. sCD40L interacts with its receptor CD40, expressed on ECs, and may lead to EC dysfunction and eventually contributes to vascular remodeling in PAH. Altogether implicating the role of platelets in EC dysfunction and thrombosis in PAH. Although there is considerable evidence to suggest that platelets contribute to the EC dysfunction and the pathogenesis of PAH, the molecular mechanisms have yet to be delineated.

### 2.4. Coagulation in PAH

Under physiological conditions, transmembrane glycoprotein TF is expressed at low levels in the pulmonary vessel wall, but its expression is significantly increased in pulmonary vascular lesions of PAH patients [61,62,63]. Increased TF/thrombin signaling contributes to vascular remodeling and the formation of plexiform lesions in PAH by inducing the proliferation and migration of SMCs and mediating the migration and angiogenesis of ECs. Furthermore, ECs from PAH patients release enhanced TF-expressing microparticles, further implicating TF as a crucial mediator in the vascular remodeling in PAH [64]. PAH patients exhibit a hypercoagulable state, consistent with the increased TF expression [65]. PAH patients have higher levels of fibrinopeptide-A (FPA), plasminogen activator inhibitor-1 (PAI), and thrombin, and lower levels of thrombomodulin [66]. Although all of the factors involved in coagulation cascade are increased in PAH, the relative contribution of EC dysfunction to their increase remains to be elucidated.

### 2.5. EC Metabolism

ECs in PAH have a metabolic phenotype that is similar to that seen in cancer. ECs in PAH have a metabolic phenotype similar to that seen in cancer, namely a metabolic reprogramming towards increased glycolytic metabolism which renders ECs with a pro-survival advantage and higher proliferation [67,68]. This metabolic shift is thought to be driven through the upregulation of glycolytic enzymes PFKFB3, hexokinase, and lactate dehydrogenase, and mitochondrial enzyme pyruvate dehydrogenase kinase (PDK) [67,69]. Therefore, the concept of targeting EC metabolism to treat PAH is emerging and raised great scientific interest. Based on a recent study in rodents, one such potential target could be PFKFB3. The blockage of endothelial PFKFB3 has shown to attenuate PH development in rats that were treated with SuHx [70]. Moreover, dichloroacetate (DCA), which is an inhibitor of the mitochondrial enzyme PDK, has been found to improve patient hemodynamics and functional capacity in genetically susceptible PAH patients [71]. Despite promising results and being based on metabolomic heterogeneity of PAH [72], comprehensive metabolic characterization of ECs still needs further investigation to further expand our understanding of the complex pathobiology of PAH.

### 2.6. Shear Stress

Abundant evidence demonstrates that shear stress is altered in the pulmonary vasculature in PAH. PAH is strongly associated with increased main pulmonary artery diameter and reduced main pulmonary artery flow rate, which suggests that the shear stress is lower globally and, thus, leads to a reduction in NO release from the endothelium [73]. Several studies found 2–3-fold lower shear stress in PAH patients when compared to control subjects, and such a reduction has a correlation with a reduction in NO bioavailability in PAH patients. This implies that the pruning of the distal pulmonary vasculature in PAH may be a way for the lung to preserve microvascular perfusion by increasing microvascular resistance and elevating shear stress [74]. However, like congenital heart disease, the microvasculature in PAH may also experience high shear stress or high oscillations in flow, sue to increased stiffness in the pulmonary arteries [75]. Despite the lower shear stress in the main pulmonary arteries, the pulsatility may elevate in the microvasculature and the stiffness of the arteries increases, which explains the coupling of microvascular dysfunction with macrovascular dysfunction.

Interestingly, decreasing the pulmonary flow via banding prevented the development of plexiform lesions in a rat model of PAH, which suggests a causative role for increased force transmission in the initiation and development of PAH [76,77]. However, pulmonary artery banding in rats induced right ventricle dysfunction [78]. Furthermore, PAH patients treated with vasodilators have shown increased survival, suggesting that dampening microvascular shear stress or pulsatile flow may improve PAH.

Using microvascular ECs derived from PAH patients, Szulcek et al. demonstrated that PAH ECs show a delayed shear adaptation and, thus, promoted shear induced endothelial dysfunction and abnormal vascular remodeling [79]. In another study, pulmonary artery ECs were subjected to high pulsatile flow, but the same mean shear stress displayed exacerbated inflammation and increased cell elongation, which could all be normalized by stabilization of microtubules [80]. Future research should focus on decoupling the microvascular shear stress, pulsatile flow, oscillation index, and right ventricular function using in vitro and in vivo models to better understand the contribution of shear stress to the EC dysfunction and development of PAH.

## 3. Features of EC Dysfunction

PAH is characterized by a dysfunctional endothelium, of which the balance between vasodilation and vasoconstriction, but also the growth factor production and cell survival are altered (Figure 3). In addition, ECs undergo endothelial to mesenchymal transition (EndoMT), which, all together, causes perturbations in pulmonary vascular homeostasis that promote vascular remodeling (Figure 4).

### 3.1. Perturbations in Vasoactivity

Reduced vasorelaxation in PAH mainly contributes to the altered expression of the vasodilators NO and prostacyclin. NO is a fast-reacting endogenous free radical that is produced by endothelial NO Synthase (eNOS). NO is essential for vasorelaxation via PASMCs, but it also has antithrombotic effects and controls EC differentiation and growth [81,82,83]. NO has long been implicated in the pathogenesis of PAH, and the lungs of PAH patients have reduced NO expression [84] (Figure 3). Whole exome sequencing has identified that mutations in Caveolin-1 are associated with PAH. Caveolin-1 is highly expressed in ECs and, interestingly, the C-terminus of caveolin-1 directly interacts with eNOS, which may result in the disruption in NO levels, ultimately triggering PAH [85]. However, other studies reported contradictory results and some PH patients even show an increase in eNOS expression [84]. Furthermore, eNOS^-/-^ mice show reduced vascular remodeling after chronic hypoxia that is caused by reduced vascular proliferation [86], pointing out the complexity of its role in PAH. Prostacyclin, also produced by EC with additional antithrombotic and antiproliferative properties [8,87,88,89], is synthesized from arachidonic acid, by prostacyclin synthase, and cyclo-oxygenase (COX) [90]. Decreased prostacyclin levels are measured in various patients with different forms of PAH, like iPAH and HIV-associated PAH [8,91], explaining, in part, the increase in pulmonary vasoconstriction, SMC proliferation, and coagulation occurring in these patients. Interestingly, in experimental PH models, mice overexpressing prostacyclin synthase are protected from developing chronic hypoxia-induced PAH [92].

ET-1, on the other hand, is a potent vasoconstrictor, which is mainly synthesized in EC and the lungs show the highest level of ET-1 in the entire body [93]. ET-1 exhibits its effects by binding to the ET_A_ and ET_B_ receptors, which activate signalling pathways in vSMCs regulating proliferation, vasorelaxation and vasoconstriction [89,93]. ET_A_ is predominantly expressed on vSMCs and is involved in vasoconstriction and proliferation of these cells, while ET_B_ is expressed on vSMCs and PAECs, and is involved in stimulating the release of vasodilators, like NO and prostacyclin, and the inhibition of apoptosis [55,89,93,94,95]. The expression of ET-1 and its receptors is increased in lungs of PAH patients and experimental PH models (Figure 3) [96,97,98,99]. Furthermore, a correlation exists between the expression of ET-1 and an increase in pulmonary resistance in PAH [98]. The increased synthesis of endothelial ET-1, accompanied with an increase in expression of ET_A_ on PASMCs, likely contributes to the increased vasoconstriction and vascular remodelling observed in PAH [88,99,100]. Another vasoconstrictor, thromboxane A_2_, which is produced by ECs and platelets, but is also an inducer of platelet aggregation and a vSMCs mitogen, is increased in PAH [8,14], creating an imbalance that might contribute to excessive platelet aggregation and vascular remodeling observed in PAH [14] (Figure 3).

At last, the expression of the growth factor vascular endothelial growth factor (VEGF) and its receptor VEGF receptor 2 (VEGFR2) are found to be increased in ECs from plexiform lesions from iPAH patients. Additionally, the plasma levels of VEGF are found to be elevated in PH patients [101,102]. The relation between PAH and increased VEGF expression is still poorly understood. It is suggested that VEGF levels in PAECs are elevated in early stages of PAH as a protective response, while, during disease progression, VEGF keeps promoting the growth of PAECs, causing the formation of plexiform lesions [8].

### 3.2. Endothelial to Mesenchymal Transition

EndoMT is a phenomenon where ECs acquire a mesenchymal-like phenotype that is accompanied with a loss of endothelial markers and increase of mesenchymal markers. In addition, ECs lose cell-cell contact, change their morphology, and adopt a highly migratory and invasive phenotype, thereby losing features of a healthy endothelium (Figure 4) [103,104]. In the lungs of human PAH patients and monocrotaline (MCT) and Sugen/hypoxia (SuHx) experimental PH rat models, EndoMT was observed, whereby cells express high levels of α-SMA and activated phospho-vimentin and VE-cadherin, indicating their endothelial origin [105,106,107]. Moreover, TWIST1, which is a key transcription factor in inducing EndoMT, is highly expressed in human PAH lungs as compared to healthy lungs [106] (Figure 4).

TGFβ treatment of PAECs induces the expression of the EndoMT transcription factors TWIST1 and SNAIL1 [103,108] and the mesenchymal markers α-SMA and phospho-vimentin [109] (Figure 4). TWIST1 increases the expression of TGFβ, leading to enhanced TGFβ signaling [110]. In addition, reduced BMPR2 signaling promotes EndoMT via the upregulation of the High Mobility Group AT-hook 1 and its target gene SLUG, independent of TGFβ signaling [111]. More interestingly, BMP-7, a protein previously described as having anti-inflammatory and anti-tumor effects in several diseases, was attenuated by hypoxia-induced EndoMT in PAECs both in vivo and in vitro by inhibiting the m-TORC1 signaling pathway [112]. BMPR2 loss favors EndoMT, allowing for cells of myo-fibroblastic character to create a vicious feed-forward process, leading to hyperactivated TGFβ signaling [113]. In summary, alterations in TGFβ/BMP signaling are linked to the process of EndoMT that was observed in PAH [114].

Hypoxia is also an inducer of EndoMT through hypoxia-inducible transcription factor-1α (HIF-1α) and HIF-2α, and both transcription factors are increased in PAH [115,116] (Figure 4). PAH ECs display an increased expression of HIF-2α, leading to SNAIL upregulation [107]. In addition, HIF-1α knockdown alone effectively blocks hypoxia-induced EndoMT, but also the knockdown of its downstream target gene TWIST1 showed the effective blockage of hypoxia-induced EndoMT in microvascular ECs (MVECs); however, it was less pronounced [117]. Nonetheless, it is important to realize that microvascular endothelium may differ from arterial endothelial function. Finally, in addition to transcription factors, microRNAs, such as miR-181b, have been shown to be implicated in EndoMT in PAH. The overexpression of miR-181b in rat pulmonary arterial ECs (rPAECs) attenuated inflammation-induced EndoMT by inhibiting the expression of TGF-βR1 and circulating proteoglycan endocan [118].

### 3.3. Apoptosis

EC apoptosis may also play a role in PH development via vascular dropout and selection pressure on ECs, contributing to the apoptosis-resistant phenotype of ECs in vascular lesions [119]. Several attempts were made in order to elucidate the molecular pathways that are involved in the regulation of PAEC apoptosis. The hypothesis is that disturbed responses to VEGF signaling, in combination with hypoxia, cause an initial increase in apoptosis in PAECs, leading to the emergence of aggressive apoptosis resistant and hyperproliferative ECs that cause the formation of intimal lesions [120,121,122]. A possible explanation for the initial increase in apoptosis of PAECs is that the loss of BMPR2 signaling promotes mitochondrial dysfunction and subsequent PAEC apoptosis [123]. White et al., interestingly, proposes a model in which the pro-apoptotic factor programmed cell death-4 (PDCD4) activates the cleavage of caspase-3, inducing PAEC apoptosis. Interestingly, they show that reducing PDCD4 levels in vivo by overexpressing miRNA-21 prevents PH development in SuHx rats [124]. Besides an initial increase in apoptosis, PAH is also characterized by PAECs that are hyperproliferative and apoptosis resistant [122]. PAECs from iPAH patients showed an increased expression of pro-survival factors IL-15, BCL-2, and Mcl-1, together with persistent activation of the pro-survival STAT3 signaling pathway [122]. Furthermore, Notch1 was elevated in lungs from iPAH patients and from SuHx rats. Notch1 contributes to PAH pathogenesis by increasing EC proliferation and inhibiting apoptosis via p21 downregulation and regulating BCL-2 and survivin expression. Furthermore, HIF1α expression promotes Notch signaling human PAECs [125]. In contrast, Miyagawa et al., demonstrated that contact-mediated communication between SMC and EC activates EC derived Notch1 and alters the cells epigenome in order to regulate Notch1-dependent genes that maintain endothelial integrity and prevent pulmonary vascular remodeling in a murine model of hypoxia-induced pulmonary hypertension [126]. Therefore, the role of Notch1 is complex and controversial in PAH and warrants more research to delineate the molecular mechanisms.

## 4. Epigenetics

In recent years, epigenetics has become a growing field of interest in PAH research. Currently, the main focus of study for targeting PAH is the following three mechanisms of epigenetic regulation: DNA methylation, histone modifications, and RNA interference (Figure 5) [17].

DNA methylation profiling of PAECs from iPAH and hPAH patients revealed differences in the expression of several genes that are involved in inflammatory processes, remodeling, and lipid metabolism when compared to the controls [127]. Among those genes, ABCA1 was found to be most differently methylated/downregulated in the discrimination between PAH and controls. ABCA1 belongs to the family of ATP binding cassette (ABC) transporters that are important for pulmonary homeostasis [127]. Furthermore, ABCA1 is linked to PAH pathophysiology in a MCT animal model of PAH, where the activation of ABCA1 improved RV hypertrophy and pulmonary haemodynamics [17,127].

Increased histone acetylation through histone-deacetylases (HDAC) is associated with vascular remodeling found in PAH [128,129]. In humans, HDAC enzymes are divided into four classes: class-1 HDACs (HDAC-1, -2, -3, and -8), class-2a HDACs (HDAC-4, -5, -7, and -9), class-2b HDACs (HDAC-6 and -10), class-3 HDACs (Sir2-like proteins), and class-4 HDACs (HDAC-11) [130]. HDAC-1 and -5 show increased expression in both lungs of iPAH patients and chronic hypoxic rats whereas HDAC-4 was only increased in human iPAH lungs [129]. More recently, HDAC-6 has been linked to PAH pathogenesis, possibly through the upregulation of HSP90 [131]. HDAC-6 was overexpressed in PAECs and PASMCs of PAH patients and PH experimental models [132]. In the SuHx and MCT rat model pharmacological HDAC-6 inhibition improved PH [132]. Several other studies showed that class-1 HDAC inhibitors attenuate PAH by suppressing arterial remodeling in a chronic hypoxia model and by reducing inflammation in PH-fibroblasts [129,133,134]. In PAECs, class-2a HDAC inhibitors restore the levels of myocyte-enhancer-factor-2 and attenuate PAH in both the MCT and SuHx PAH rat models [135].

The epigenetic regulator bromodomain-containing-protein-4 (BRD4) is linked to the pathogenesis of PAH [136]. BRD4 is a member of the Bromodomain and Extra-Terminal (BET) motif family, which binds histones to influence gene expression [137]. BRD4 is overexpressed in the lungs of PAH patients in a miR-204 dependent manner. It inhibits apoptosis by sending cell survival signals [136,138], and stimulates the proliferation of PAEC and PASMC proliferation at these sites [17,138]. The selective inhibition of BRD4 with RVX-208 restored EC function, reversed PAH in the MCT and SuHx rat models, and supported the RV function in pulmonary artery banding model of PAH [136].

## 5. EC Dysfunction in Other PH Groups

Patients with PAH, which are classified as group 1, are just a proportion of the five broad groups of patients suffering from PH. The remaining groups, group 2 (PH due to left-sided heart disease), group 3 (PH due to lung disease), group 4 (PH due to chronic thromboembolic disease), and group 5 (PH due to unclear and/or multifactorial mechanisms), also present signs of EC dysfunction.

### 5.1. Group 2 PH

Group 2 PH is due to a complication of left heart disease and it is most common in patients with heart failure (HF). Therefore, research in group 2 PH mostly focuses on left ventricular dysfunction and not so much the lung vasculature. However, features of EC dysfunction are also observed in PH-LHD [139]. An experimental model of chronic HF showed reduced NO activity and responsiveness to NO in pulmonary arteries [140]. Moreover, ET-1 is elevated in certain PH-LHD phenotypes and ET-1 activity is increased in plasma of patients with chronic HF. Blocking the ET_A_ receptor caused pulmonary vasodilation in these patients [141,142]. Furthermore, polymorphisms that are found in eNOS also contribute to PH development in patients with LHD [143]. Despite the presence of similar perturbations in vasoactivity between PAH and PH-LHD, treating PH-LHD patients with drugs used to treat PAH patients was not beneficial and even harmful [139,144].

### 5.2. Group 3 PH

Chronic obstructive lung disease (COPD) associated PH is the best described form of PH in group 3. The main trigger of COPD is considered to be cigarette smoke, which causes chronic inflammation in the lung that subsequently triggers EC dysfunction and leads to PH [145]. Cigarette smoke decreases eNOS and prostacyclin expression in PAECs [146,147]. COPD patients can show the overexpression of VEGF and ET-1 in pulmonary arteries [148,149]. A role for HIF1α and EndoMT has also been suggested in COPD [150,151]. Although there are similarities in EC dysfunction, the drugs used to treat PAH are currently not recommended for group 3 PH, due to a lack of evidence how these drugs may influence PH progression in combination with the underlying lung diseases [152].

### 5.3. Group 4 PH

CTEPH develops as a result of a pulmonary embolism (PE) that does not resolve [153]. These organized pulmonary thrombi in the lungs are associated with distal vascular remodeling of non-occluded vessels similar to the remodeling observed in PAH lungs [153]. Whether patients develop CTEPH due to primary EC dysfunction or as a consequence of PE remains to be resolved. Nevertheless, evidence supports that features of EC dysfunction, which are similar to those observed in PAH, are present in these patients and could play a causal role in CTEPH development. Activated platelets with a hyper-responsiveness to thrombin are likely to contribute to the CTEPH pathogenesis and progression via enhancing inflammatory responses of pulmonary ECs [154]. EC dysfunction-associated vascular remodeling has been suggested as a common mechanism between CTEPH and PAH [153,155]. Primary cell cultures that were isolated from endarterectomized tissue co-expressed both EC and SMC markers, suggesting a role for EndoMT in intimal remodeling/lesion development in CTEPH [156]. The existence of endothelial dysfunction in CTEPH pathogenesis is further supported by the fact that conditioned medium from CTEPH-derived PAECs, containing high levels of growth factors and inflammatory cytokines, increased PASMC proliferation and monocyte migration [157]. In addition, PAECs from CTEPH patients show an increased proliferation, altered angiogenic potential and metabolism, and apoptosis resistance [158,159,160,161,162]. Increased levels of soluble intracellular adhesion molecule-1 (ICAM1) in PAECs from CTEPH patients and in endarterectomy may contribute to EC proliferation and apoptosis resistance through its effect on cell survival pathways [161]. Additionally, FoxO1, in a PI3K/Akt dependent manner, is a possible contributor to the loss of balance between cell survival and death and it was downregulated after PE in a rat model of CTEPH [163]. A recent study reported that decreased levels of ADAMTS13 and increased levels of vWF levels were observed in plasma of CTEPH patients, suggesting the role of the ADAMTS13–vWF axis in CTEPH pathobiology. However, it remains unknown as to whether this axis plays a role in EC dysfunction in CTEPH [164]. Finally, PAECs isolated from CTEPH patients showed a significant rise in basal calcium levels, which is an important regulatory molecule for EC function [165]. This imbalance in calcium homeostasis is caused by angiostatic factors, such as PF4, IP-10, and collagen type 1, which are formed in the microenvironment that is created by the unresolved clot and eventually leads to EC dysfunction [165]. So far, a soluble guanylate cyclase stimulator (Riociguat) is the only PAH based therapy that has been approved in patients with CTEPH that are not eligible for surgery [166].

## 6. Current and Future Perspectives

Although much progress has been made to understand EC dysfunction in PAH, to date there is still no definitive cure and patients only have a median survival rate of 2.8 years [167]. Current therapies for PAH, which consist of calcium channel blockers, ET-1 receptor antagonists, phosphodiesterase type 5 inhibitors, prostacyclin-derivatives, and, more recently, also Riociguat, focus on restoring the imbalanced endothelial vasoactive factor production to promote SMC relaxation, but with limited or no effect on other features of EC dysfunction and subsequent progressive pulmonary vascular remodeling [168,169,170]. Therefore, research on EC dysfunction and its stimuli to target structural changes that narrow lumen size in PAH is vital to find a cure.

A first step towards reversing vascular remodeling in PAH is the use of apoptosis-inducing drugs, such as anthracyclines and proteasome inhibitors. They are already used in combination with cardio-protectants, such as p53 inhibitors, to reduce pulmonary pressure and restore blood flow in the experimental models of PAH [171,172]. The combinatorial use is essential in circumventing the lack of cell-type/organ specificity of cell-killing drugs. Cancer patients, but also experimental PAH animals treated with only cell-killing drugs, show signs of cardiotoxicity that should be prevented in PAH patients that already suffer from reduced right heart function [171,173,174,175].

Another way to target progressive pulmonary vascular remodeling focuses on restoring signaling pathways and EC function, e.g., using selective TGF-β ligand traps [176] or TGF-β synthesis inhibitors, like kallistatin, which are known to improve hemodynamics, remodeling, and survival in experimental PH models, and to inhibit EndoMT in HUVECs, stimulate eNOS expression, and prevent TGF-β induced miRNA-21 synthesis, respectively [176,177]. However, blocking inflammation to restore normal EC function in PAH was not successful. One explanation might be the complexity of the immune system and, by inhibiting the bad side, one also suppresses beneficial inflammatory pathways [178,179].

Modulating BMPR2 has also been proposed as a therapeutic approach to reverse endothelial dysfunction in PAH. A recent study comparing human induced pluripotent stem cell-derived ECs (iPSC-ECs) from unaffected BMPR2-mutation carriers with iPSC-ECs from BMPR2-mutation carriers that present PAH identified several BMPR2 modifiers and differentially expressed genes in unaffected iPSC-ECs. These BMPR2 modifiers exert a protective response against PAH by improving downstream signaling, which compensates against BMPR2 mutation-induced EC dysfunction and offers insights towards new strategies for rescuing BMPR2 signaling [180]. A potential therapy for stimulating BMPR2 signaling is through pharmaceuticals [181]. Direct enhancement of endothelial BMPR2 signaling using recombinant BMP9 protein prevents and reverses the established experimental PAH [42]. However, in contrast to Long et al., Tu et al. (2019) showed that the deletion or inhibition of BMP9 protects against experimental PH via its effect on endothelial production of ET-1, apelin, and adrenomedullin [182]. In line with this, we have recently shown that BMP9-induced aberrant EndoMT in PAH pulmonary ECs is dependent on exacerbated pro-inflammatory signaling mediated through IL6 [54]. These studies show the BMP receptor family complexity as therapeutics in PAH. More recently, ACTRIIA-Fc, an activin and growth and differentiation factor (GDF) ligand trap, prevented and reversed existing PH in experimental PAH models. ACTRIIA-Fc inhibited SMAD2/3 activation and restored a favorable balance of BMP signaling versus TGF-β/activin/GDF signaling. ACTRIIA-Fc is currently tested in a phase-2 clinical trial for efficacy and safety in PAH patients (NCT03496207) [183]. However, a recent study shows that TGF-β/SMAD signaling is regulated differently in PH animal models compared to PAH patients [184]. Therefore, more research should be performed on this complex TGF-β/activin/GDF signaling. Spiekerkoetter et al. uncovered a molecular mechanism, where FK506 (tacrolimus) restores defective BMPR2 signaling in PAECs from iPAH patients and reverses severe PAH in several rat models [181]. Based on improvements in clinical parameters and the stabilization of cardiac function of end-stage PAH patients in a phase-2a clinical trial, a low dose of FK506 was proposed as potentially beneficial in the treatment of end-stage PAH [185]. These findings open-up an area, in which correcting BMPR2 mutations in combination with other therapies might be more successful in curing PAH. A proposed hypothesis to cure PAH describes collecting iPSCs from PAH patients, restoring the BMPR2 mutation with CRISPR/Cas9, and reinjecting those iPSCs in the patient to normalize EC function and signaling along with administration of drugs that could restore the protective gene expression profile of unaffected BMPR2 mutation carriers [186]. 6-Mercaptopurine (MP), which is a well-established immunosuppressive drug, inhibits EC dysfunction and reverses development of PH in the SuHx rat model by restoring BMP signaling through the upregulation of nuclear receptor Nur77 [187]. A recent proof-of-concept study with MP in a small group of PAH patients showed a significant reduction pulmonary vascular resistance, accompanied by increased BMPR2 mRNA expression in the patients’ peripheral blood mononuclear cells. However, unexpected severe side-effects require further dose optimization and/or the use of other thiopurine analogues [188]. Next to a role for BMPR2, the loss of KCNK3 function/expression is a hallmark of PAH. A recent study shows that the loss of KCNK3 is inducing EC dysfunction by promoting the metabolic shift and apoptosis resistance in PAECs. Therefore, targeting KCNK3 might restore EC function; however, the mechanisms remain unknown [189]. The transplantation of mesenchymal cells in rats from the SuHx model improved haemodynamic parameters, but, more interestingly, reduced EndoMT (partially) through the modulation of HIF2α expression [190]. Furthermore, mesenchymal stem cells are also suggested to reduce inflammation through the secretion of paracrine factors and attenuate vascular remodeling by lowering collagen deposition [190,191,192]. However, the underlying mechanisms for this observation remain unclear [190]. Several recent studies demonstrate the role of endothelial HIF-2a in the pathogenesis of PAH, and therapeutic targeting of HIF-2a with small molecule inhibitors, such as PT2567, have showed a beneficial effect in PAH in vivo [193,194]. A recent study demonstrates that human pulmonary ECs of patients with PAH are more vulnerable to cellular senescence, a process that is associated with EC dysfunction. Interestingly, targeting senescence while using the senolytic drug ABT 263 reversed established PH in a MCT+shunt induced PAH rat model by specifically inhibiting senescent vascular cells [77]. However, more research should be performed on the safety and efficacy of senolytics in patients.

Finally, epigenetic modulation has received growing interest as potential therapeutic intervention. Especially, specific HDAC inhibition shows great promise in reversing pulmonary remodeling and pressure [129]. A problem with broad-spectrum HDAC drugs is that they show severe side effects on the right ventricle, which can have fatal consequences in PAH patients with RV failure [133,195,196]. Therefore, searches for more selective HDAC inhibitors that do not show cardiotoxicity are still being done. One example is MGCD0103, which is a HDAC inhibitor that selectively inhibits class-1 HDACs that has been tested in a chronic hypoxia rat model. This inhibitor showed improved hemodynamics, reduced wall thickening, while RV function was maintained [133]. Additionally, BET inhibitors, such as RVX208, seem to be promising in the treatment of PAH through its beneficial effect on reducing the apoptosis-resistant and pro-inflammatory phenotype in PASMCs and MVECs isolated from PAH patients, but also on vascular remodeling and the RV in several experimental models of PH [136]. Finally, miRNA-21 has been associated with multiple pathogenic features, such as TGF-β signalling, EndoMT, and apoptosis, which are central to PAH. Therefore, therapeutic modulation of miRNA-21 may be an important issue for future research to restore pathogenic signaling.

## 7. Conclusions

To date, we still do not fully understand what triggers the onset and progression of PAH. We do know that BMPR2 mutations, epigenetics, physiological conditions, and inflammation are important triggers. EC dysfunction plays a central role in all of this, through EC proliferation, EndoMT, and a misbalanced production of vasoactive factors, resulting in the disorganized growth of PASMCs. Although several preclinical studies demonstrate that EC dysfunction is a cause rather than a consequence of PAH, more research should be performed in PAH patients in order to better understand this. For example, non-carriers of BMPR2 mutation along with carriers of BMPR2 mutation from the same family should be followed up for several years to understand whether EC dysfunction or other triggers are a cause or consequence. Despite advancements that have been made in treating this disease, very few therapies have little or no direct impact on EC dysfunction. Therefore, successful treatments should focus on multiple aspects of EC dysfunction and not solely on its effect on SMCs and fibroblasts in PAH. A better understanding of the molecular mechanisms that are involved in EC dysfunction in PAH is of utmost importance for developing successful therapies to save the lung as well as the heart, and perhaps cure PAH in the future.

## Figures and Tables

**Figure 1 biomedicines-09-00057-f001:**
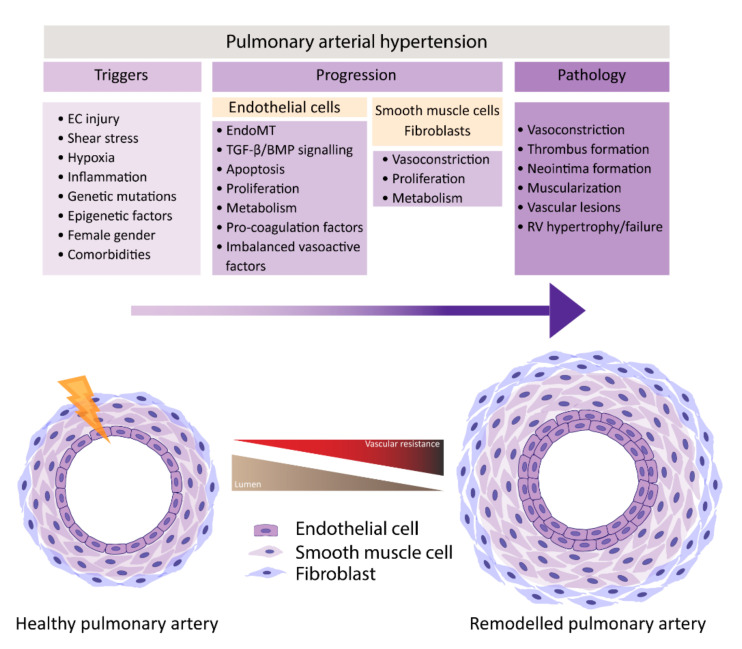
Pulmonary artery remodeling, vascular resistance and pulmonary arterial hypertension (PAH) development. PAH results from a progressive increase in vascular resistance caused by pulmonary vascular remodeling. Molecular mechanisms behind the process of vascular remodeling are still not fully elucidated but endothelial cell (EC) injury is thought to be one of the early triggers. EC injury can be caused by shear stress, hypoxia and inflammation. Host factors such as genetic mutations and gender but also epigenetic factors and comorbidities are thought to play an important role in EC dysfunction. EC dysfunction leads to altered cell signaling that induces cellular processes such as EndoMT, apoptosis, and proliferation. In addition, changes are found in cell metabolism and in the secretion of vasoactive, coagulation and thrombotic factors. Additionally, vascular smooth muscle cells and fibroblasts are found to display a diseased cellular phenotype. EC dysfunction eventually promotes vasoconstriction, thrombus formation, neointima formation, muscularization, and the development of vascular lesions. As lumen size decreases, pulmonary vascular resistance increases and induces right ventricle (RV) hypertrophy, with eventual RV failure.

**Figure 2 biomedicines-09-00057-f002:**
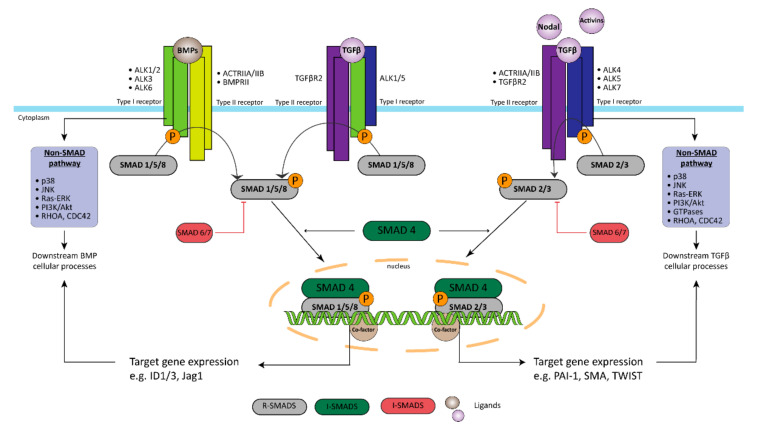
Transforming growth factor-β (TGF-β) superfamily signaling in PAH. The bone morphogenetic protein (BMP)/TGF-β signaling pathway is an important factor in the existence of EC dysfunction in PAH. Decreased expression of BMPR2 but more importantly various mutations in the BMPR2/BMP-ligand axis are associated with specific changes in EC behavior such as increased proliferation and migration, but also structural changes that cause the loss of the protective EC barrier. In addition, the TGF-β superfamily signaling also plays an important role the initiation of EndoMT by triggering the overexpression of genes, like TWIST1, αSMA, and phospho-vimentin. Receptor-regulated Smads (R-Smads); Common mediator Smad (Co-Smad); Inhibitory Smads (I-Smads).

**Figure 3 biomedicines-09-00057-f003:**
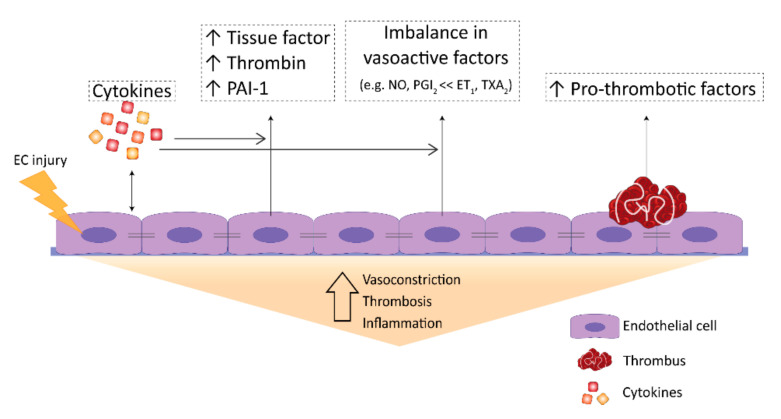
Endothelial dysfunction in PAH. PAH is characterized by endothelial dysfunction that causes an imbalance in the production of several endothelial-specific factors. The endothelium presents a pro-inflammatory phenotype with an increased expression of cytokines, a pro-thrombotic surface due to changes in the expression of clotting factors (e.g., TF) and increased expression of pro-thrombotic factors, and an imbalanced production of vasoactive factors that promote vasoconstriction. Upon endothelial cell injury, pulmonary artery- endothelial cells (PAECs) become dysfunctional and alter their secretion of cytokines and other factors that regulate coagulation, thrombosis, and vascular tone. A failure of PAECs in maintaining vessel homeostasis promotes vasoconstriction, thrombosis and inflammation that initiate PAH disease progression.

**Figure 4 biomedicines-09-00057-f004:**
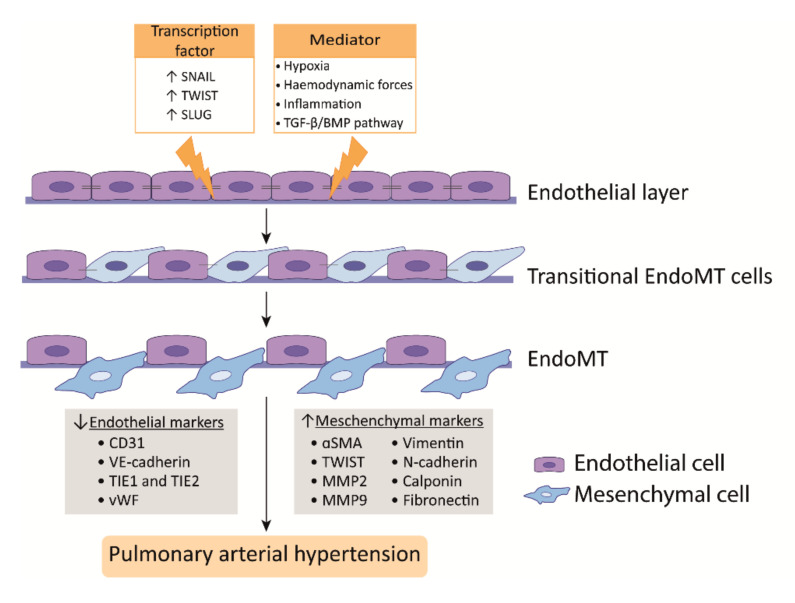
Endothelial to mesenchymal transition (EndoMT) in PAH. EndoMT in PAH is thought to be an important process contributing to vascular remodeling. Activation by transcriptional factors, hypoxia, haemodynamic forces, inflammation, and TGF-β/BMP pathway signaling pulmonary endothelial cells (PAECs) undergo cellular transition to a mesenchymal phenotype, in which PAECs lose endothelial markers and gain mesenchymal markers, such as αSMA and TWIST. These mesenchymal-like cells present an invasive character and hence contribute to vascular remodeling in PAH.

**Figure 5 biomedicines-09-00057-f005:**
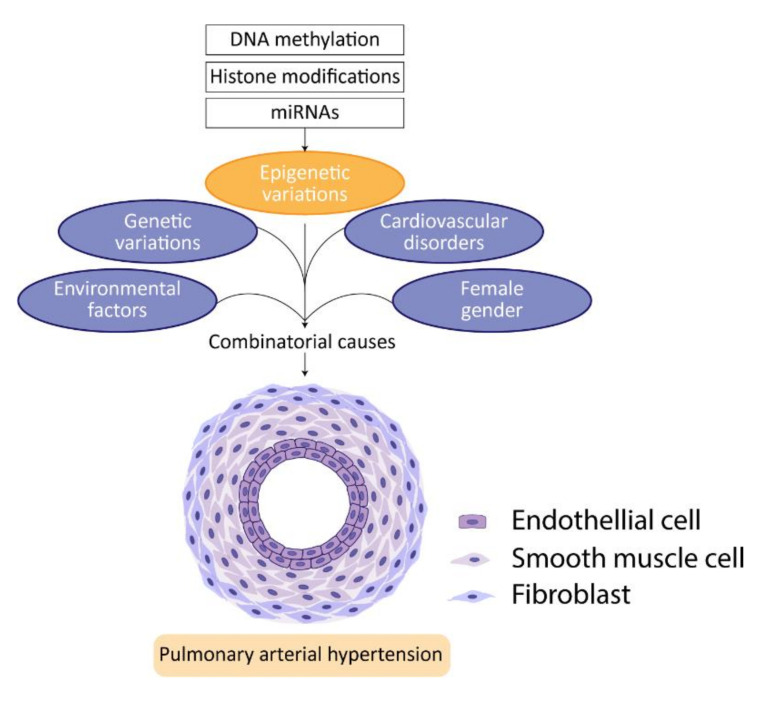
Epigenetics in PAH. In addition to genetic variations and other risk factors, such as gender, comorbidities, and environmental factors, epigenetic variations in PAH gain interest. Differences in DNA methylation profiles, increased histone acetylation and dysregulated miRNA expression in PAH patients point out a growing field in PAH research that provides better understanding of disease pathology.

## Data Availability

Not applicable.

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
