# Peer review of "Endothelial Dysfunction in Pulmonary Hypertension: Cause or Consequence?"

_biomedicines, 2021, doi:10.3390/biomedicines9010057_

Round 1
Reviewer 1 Report
Kurakula et al. reviewed endothelial dysfunction in pulmonary hypertension. Authors well summarized the latest findings related to endothelial dysfunction from a lot of studies in animal and cells. Figures are clear and are easy to understand the latest findings. I have some questions to authors.
- Are there any relation between metabolic function and endothelial dysfunction in pulmonary hypertension?
- Authors should describe the relation between shear stress and endothelial dysfunction more.
- Do authors think that endothelial dysfunction is cause or consequence? Please state your opinion.
Reviewer 2 Report
Type of manuscript: Review
Manuscript ID: 1031781
Title: Endothelial Dysfunction in Pulmonary Hypertension: Cause or
Consequence?
Authors: Kondababu Kurakula, Valerie F.E.D. Smolders, Olga Tura-Ceide, J.
Wouter Jukema, Paul H. A. Quax, Marie-José Goumans *
In this review entitled “Endothelial Dysfunction in Pulmonary Hypertension: Cause or Consequence?” Kondababu and colleagues described the factors that are related to EC dysfunction in PAH. In this review, authors outline the latest advances on the role of EC dysfunction in PAH and other forms of pulmonary hypertension focus in the endothelial to mesenchymal transition, inflammation, apoptosis, and thrombus formation. Moreover, authors described molecular signals that orchestrate EC dysfunction in PAH and the therapeutic potential of targeting this process in PAH.
In general, this is a very interesting and well-written review in the field of PAH associated endothelial dysfunction. All the sections of the review and all the information is very complete and toughly referenced . This review is certainly interesting for the readership of Biomedicines.
Some minor points should be re-evaluated and would improve the review.
MINNOR POINT;
1) I consider that, in section “2.1. Bone morphogenic type 2 receptor” is important to dedicate a deeper explanation about BMPs proteins (structure, family, members) for better understanding.
2) Authors say that mutations in the bone morphogenic type 2 receptor (BMPR2) is associated with approximately 80% of familial PAH (hPAH) and 20% of idiopathic cases of PAH (iPAH). However the low penetrance of the mutation suggest a secondary stimulus such as inflamation and coagulation.
In this sense, as inflammation is strongly implicated, and BMPR2 genetic-independed regulation as well, I recommended to refer Hurst et al 2017 Nat Commu to support inflammation as a cause of the BMP type II decreased. In this article they decribed that TNFa could triggers PAH by suppresing BMP type II receptor altering NOTCH signalling.
3) Authors describe the phenomenon of EndoMT carefully in PAH. However, I suggest that would be interesting to include the article Zhao et al. (2020) in which they described mechanistically that overexpression of miR-181b in rat pulmonary arterial ECs (rPAECs) inhibit TNF-α, TGF-β1, and IL-1β-induced EndMT by inhibiting the expression of TGF-βR1 and circulating proteoglycan endocan.
Reviewer 3 Report
This is a well-organized review in the topics of endothelial dysfunction in pulmonary hypertension, which has attracted a lot of attention in the field of PH. However, some important aspects related to EC are not included or not covered, which limits the enthusiasm of this review. Please see the following comments. Although recombinant protein BMP9 treatment enhanced EC BMPR2 signaling and reversed PH in rodent models, there was another study with opposite data showing that BMP9 -/- or anti-BMP9 antibodies prevent hypoxia-induced PH, suggesting that the molecular mechanisms of BMP ligand and receptor involved in PAH are complex and need to do further studies. (PMID: 30636542) Many pieces of evidence were demonstrating that inflammation is one of the key triggers to the development of PAH, for example, PAH associate with autoimmune disease, PAH associated with HIV. This part should be revised to include this information. The section on Thrombosis and coagulation does not include useful information. Very little literature was covered. This section should be revised and cover. When NO/eNOS was raised, eNOS binding partner Cav1, which is found mutated in PAH patients and highly expressed in ECs, was not introduced and discussed. Page 8, Line 270. The role of Notch1 in EC in PAH is complex and controversial. Another study demonstrated that deletion of EC-Notch1 in transgenic mice worsens hypoxia-induced pulmonary hypertension. This paper should be mentioned. PMID: 30582451 The study on BRD4 is on PASMCs. Its role in EC remains unclear. It is better to remove this part. ECs secrete many growth factors and cytokines and exhibit the autocrine and paracrine effect. Several studies have demonstrated that EC derived factors including FGF2, PDGFB, CXCL12 affect other perivascular cell behavior including smooth muscle cells, fibroblast, and pericytes. PMID: 29664678, PMID: 21037114. Recent studies have demonstrated the important role of endothelial HIF-2a in the pathogenesis of PAH, and therapeutic targeting HIF-2a with small molecules have showed the promising value in preclinic studies. PMID: 29924941, PMID: 31515405, PMID: 32972983Author Response
Please see the attachment

Round 2
Reviewer 3 Report
All my concerns were addressed. Thank you for writing this awesome review.